# Precipitation Forecasting and Monitoring in Degraded Land: A Study Case in Zaghouan

Okba Weslati [1,*] , Moncef Bouaziz [2] and Mohamed-Moncef Serbaji [1]

[1] Laboratory of Water, Energy, Environment, National Engineering School of Sfax, Road of Sokra 3 km, Sfax 3038, Tunisia
[2] Institute for Mine Surveying and Geodesy, Freiberg University of Technology, 0959 Freiberg, Germany
* Correspondence: okba.weslati@gmail.com

**Abstract:** The study aimed to forecast and monitor drought over degraded land based on monthly precipitation using the Seasonal Autoregressive Integrated Moving Average (SARIMA) approach. Several statistical parameters to select the most appropriate model were applied. The results indicate that the SARIMA (1,1,1) (0,1,1)12 is the most suitable for 1981 to 2019 CHIRPS time-series data. The combination of precipitation data and this approved model will subsequently be applied to compute, assess, and predict the severity of drought in the study area. The forecasting performance of the generated SARIMA model was evaluated according to the mean absolute percentage error (15%), which indicated that the proposed model showed high performance in forecasting drought. The forecasting trends showed adequate results, fitting well with the historical tendencies of drought events.

**Keywords:** drought; forecasting; monitoring; SARIMA; precipitation

## 1. Introduction

Precipitation is considered a main component in the water cycle. The global average precipitation is estimated between 2.5 and 2.8 mm/day [1,2]. However, the distribution of rainfall is uneven, with arid and semi-arid regions suffering the most from irregular rainfalls and water scarcity, leading to drought events that affect the land. Drought can vary from year to year.

Drought is considered to be a major factor in land degradation processes. Several research works have highlighted the impacts of arid episodes and the occurrence of major degradation locally and globally [3–6]. Scientific discussion on these issues is ongoing. Scientists have identified several types of droughts as an effective way to monitor and classify them. Meteorological drought typically occurs when a dry weather pattern dominates a specific region. This type of drought is based on the degree of rainfall deficit and the duration of the dry period. It is identified when rainfall is less than average over a significant period, often a month. Hydrological drought happens when low water supplies become evident in the water system, influenced by the impact of rainfall deficits on the water system and supply, such as reservoir and lake levels, stream flow, and ground water table decreases.

Moreover, when crops are damaged and affected by drought, we define it as agricultural drought. This refers to the impacts on agriculture by various water irrigation factors (soil water deficits, rainfall deficits, ground water decline, or reservoir levels). The best way to monitor agricultural drought is by mapping rainfall records and vegetation conditions. Ecological drought happens when natural ecosystems are influenced by drought, and socioeconomic drought is felt when multiple demands and supplies of commodities (e.g., vegetables, grains, fruits and meat) are affected by drought conditions (agricultural, meteorological or hydrological drought). This generally occurs when the

demand for an economic good surpasses supply as a result of deficit in water supply caused by weather.

Many research studies have suggested a potential rise in global average temperature, which will alter climate components such as rain, snow, clouds, and the water cycle. The increase in average heat will accelerate the rate of evaporation, and more water vapor will be generated in the air causing greater rainfall. Many predictive models have indicated a probable increase of 1 °C in the average global temperature leading to a rise of 1 to 3% in yearly precipitation, reaching 12% in 2100. Mediterranean regions will get the least amount of precipitation and will be affected by a severe drought and water shortage [7]. This potential increase will lead to more snow and a higher risk of flooding, as well as other severe weather disasters. Therefore, the instability of precipitation has elicited a proactive approach for water resource management to reduce the impacts of future drought. In that context, machine learning prediction models can play an essential role in water resource management to reduce the impacts of water scarcity and climate change.

The advance of science has led to the application of artificial intelligence and machine learning for weather forecasting. Former traditional methods were based on the physical simulation of the atmosphere presuming it to be a fluid. Therefore, the approach applies thermodynamic and fluid equations to compute the future behaviour of the weather. The generated model is unstable and too sensitive because of uncertainties in measurement. Moreover, the complex, non-linear and stochastic features of climate result in a physical model based on ordinary differential equations being unable to fully represent the complex process of the atmosphere, generating unreliable forecasting models. Machine learning algorithms, on the other hand, are robust and do not require studying all the physical components that interfere with the atmosphere.

Various machine learning methods have so far been conducted in weather forecasting [8,9], suggesting new methods to improve time series predictions and accuracy improvements. The main objective of machine learning is to improve prediction accuracy by minimizing forecasting error. On the other hand, machine learning forecasting can often generate implausible solutions leading to an exaggerated simulation scenario. Therefore, the application of a certain model must be contingent on various and complex parameters before accepting the forecasting results [10].

Seasonality is one of the most important features for many time-series data. As a result, it has been found that the Seasonal Autoregressive Integrated Moving Average (SARIMA) is very effective in forecasting seasonal time series data in various fields [11–15]. The popularity of SARIMA arises from Box and Jenkin (1976) [16]. The applied methodology, while building the model, requires that all the data are seasonally differentiated to fulfill the stationarity condition. This step is required to eliminate the effect of seasonal variation which can obstruct the measurement of other time series components. As a result, the first step in this model consists of applying seasonal adjustments to remove seasonal variation. Once this is done, the model is calibrated and scaled back using specific seasonal parameters.

Many studies have widely used SARIMA models for understanding and forecasting climate variables. Dimri et al. (2020) [17] applied a combination of ARIMA and SARIMA for forecasting temperature and precipitation for the Bhagirathi River basin (India). They used multiple SARIMA versions depending on the weather component. The resulting forecasting information serves for better water management of the area. Eni and Adeyeye (2015) [18] used the diverse parameters of the SARIMA model (Akaike's Information Criterion (AIC), Schwartz's Bayesian Criterion (SBC),complemented with autocorrelation interpretation to determine the most efficient SARIMA model to fit the historical rainfall data. This model was applied to visualize and forecast precipitation amounts for 2013 in the regions of Warri town (Nigeria). Mohan and Vedula (1995) [19] used the SARIMA model to forecast monthly reservoir inflows for 27 years. The model was based on 25 years of historical data with logarithmic transformation. The results based

on the applied machine learning model proved that SARIMA is efficient in forecasting reservoir inflows.

In this paper, we applied the SARIMA model to monthly in situ precipitation data from 1981 to 2019. To identify the most appropriate SARIMA model, we first removed the seasonal variation from the historical time series data. The chosen model must meet all SARIMA statistical criteria in order to demonstrate its performance compared to other proposed models, and the selected model be used to predict precipitation behavior, where the forecast values are compared to the observed values. In the following section, we describe the study area and the type of data used for the model. We then discuss SARIMA and the best tools for validating and selecting the best model. The results and discussion section regarding the experimental setup is followed by the conclusion.

## 2. Study Area

Zaghouan region is located in the northeast of Tunisia (Figure 1), at 36°24′ North/10°09′ East. The city is situated on a hill of the Mountain of Zaghouan with an altitude of 1295 m. The region is tectonically active, known by the famous fault of Zaghouan, which extends for 80 km long in a northwest-southeast direction. The total vertical displacement of this mechanical structure is approximately 5 km. The active seismicity of the area has generated a large number of water sources. The area is also known for the abundance of hydrothermal water sources, which has made the region a famous destination for health tourism. The local climate is semi-arid, the annual average temperature is around 18 °C and the total yearly rainfall rarely exceeds 500 mm. The estimated population of the city is 20,837. Besides health tourism, a big part of local activities is based on agriculture, where almost 1.4 million quintals are produced yearly from 300,000 ha of land [20,21]. The area is suffering from land degradation (mainly soil salinization), which constitutes a threat to the safe use of the groundwater in irrigation and for drinking purposes [20,22]. The expansion of salt-induced soil in agriculture land use had intensified the desertification phenomena and crop yield loss. The area is also suffering from landscape degradation by the effect of soil erosion. The impact is amplified by the aggressiveness of the semi-arid climate, the irregularity (frequency and intensity) of rainfall, and the increased intensity of run-off water, causing multiple regions of flood-prone land. The occurrence of such phenomena has caused significant losses in soil and runoff water pouring into the seas [23,24].

The time-series variation of rainfall for the city of Zaghouan, illustrated in Figure 2, shows that precipitation has a sinusoidal trend, where maximum rainfalls occur in November, December, January, and March, and minimum rainfall is recorded in June, July, and August. This data series has a seasonal variation, where maximum precipitation is recorded in winter while the minimum rainfall occurs in summer. This climatic behaviour is considered one of the most important features that define the arid/semi-arid climate, where 50% of annual precipitation occurs during the cold season (winter). The hotter season (summer) is characterized by a high temperature and scanty rainfall.

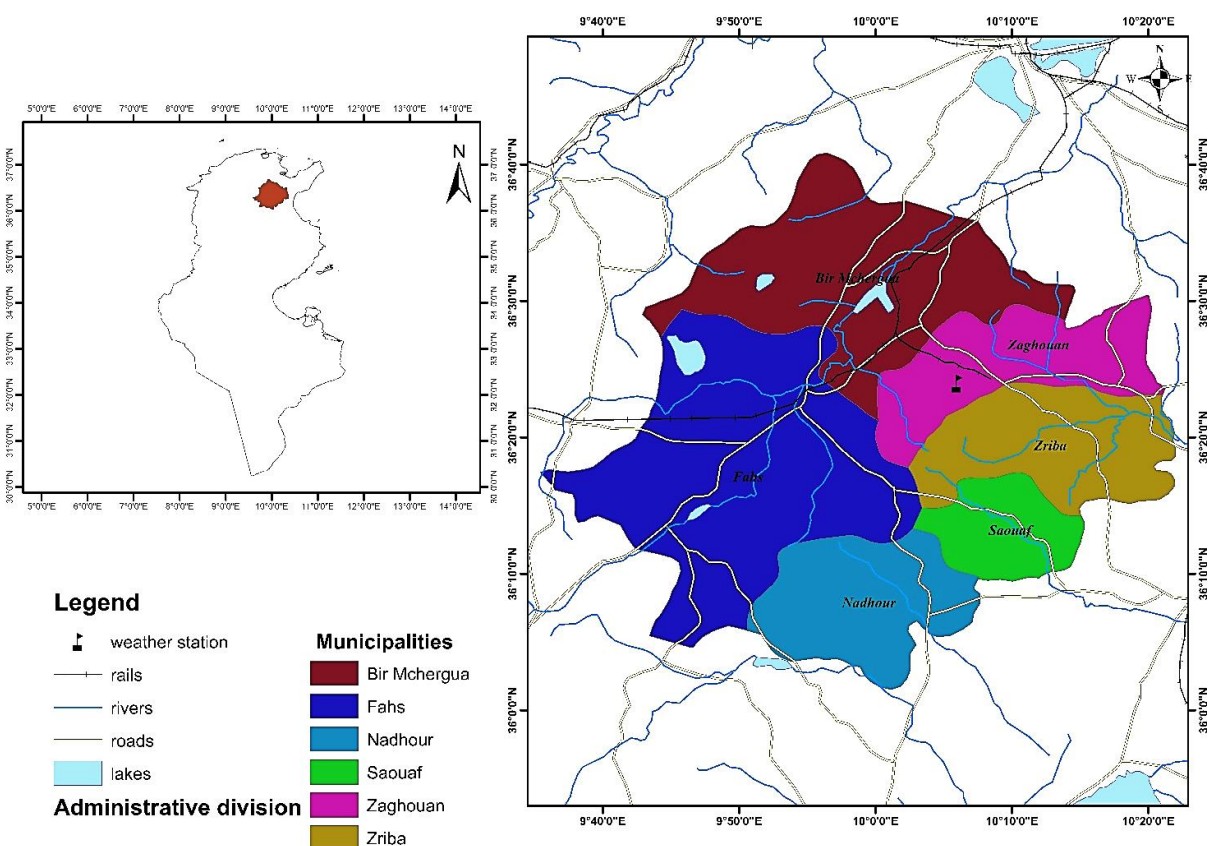

**Figure 1.** Location of the study area.

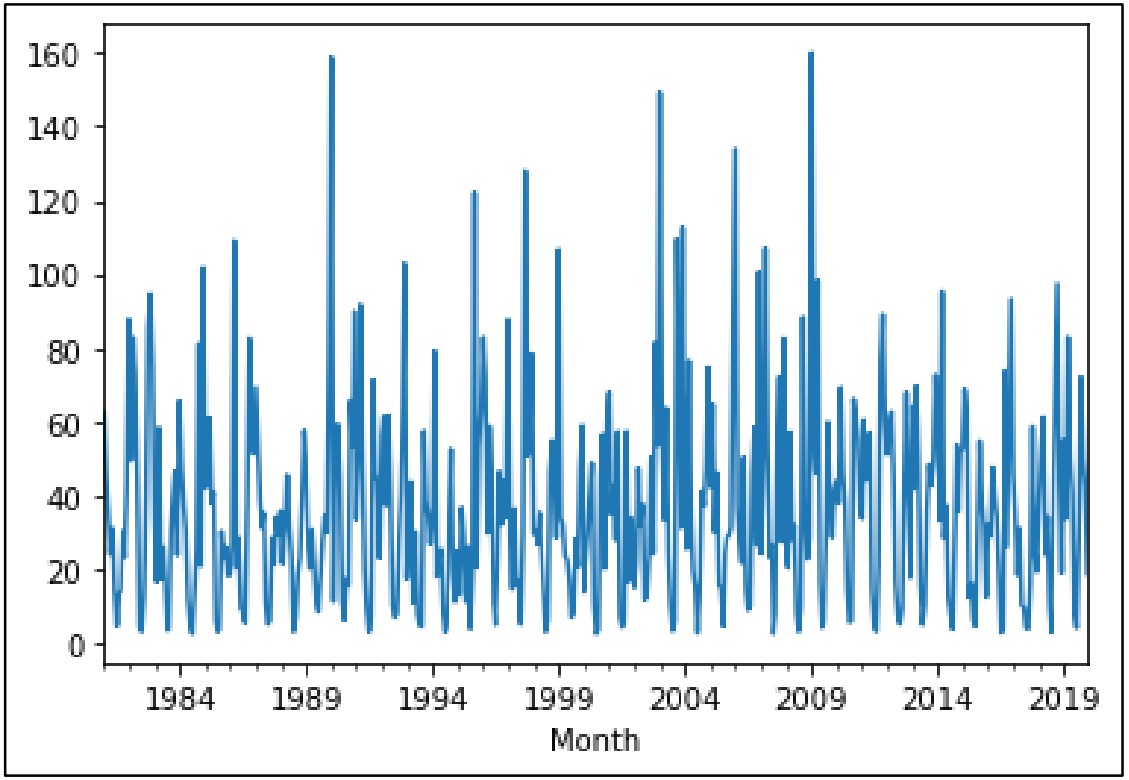

**Figure 2.** Historical monthly rainfall in the city of Zaghouan from 1981 to 2019.

## 3. Methodology

### 3.1. Data

The monthly rainfall data used in this study are Chirps precipitation data covering the period from 1981 to 2019. The data consist of 468 historical records collected from rain station located at coordinates 36°24′32.828″ N 10°8′32.341″ E in the city of Zaghouan. The collected data were used to assess the severity of rainfall deficits in Tunisia, a country with high water stress [25]. The data were simulated using advanced models of terrain-induced precipitation enhancement and compared with real station data and satellite observations. This technique, developed by NOAA and NASA, was designed to generate rainfall maps for regions with sparse surface data. With the advancement of new techniques and remote sensing observations, the goals have reached a new level, such as providing yearly warnings for trend analysis and seasonal drought monitoring. The site is now capable of providing complete, reliable, and up-to-date data sets [26].

Estimating rainfall variations (in both space and time) is a crucial aspect of environmental monitoring and drought early warning. Seasonal observation and monitoring data must be placed in a historical context to evaluate the severity of rainfall deficits. However, satellite data can now provide real and continuous averages that suffer from biases due to the complex topography, which often miscalculate the intensity of extreme precipitation events. Conversely, precipitation grids produced on the basis of station data suffer more in rural areas where there are often few (sometime no) rain-gauge stations. In that context, The Climate Hazards group Infrared Precipitation with Stations (CHIRPS) dataset is based on previous approaches to smart interpolates, with high resolution, and a long period of precipitation trends based on infrared Cold Cloud Duration (CCD) observations. The algorithm is made around a 0.05° climatology that integrates satellite information to represent sparsely gauged locations which incorporate daily and monthly precipitation estimates that extend from 1981 to the present date. The algorithms blend station data to generate a preliminary information product with a latency of approximate 2 days with an ultimate final product with an average of 3 weeks latency. The model uses a novel blending procedure that incorporates the spatial correlation structure of CCD-estimates to interpolate weights. The CHIRPS algorithm has proven the capacity of quantifying the hydrologic impacts of decreasing precipitation and increasing air temperature. Moreover, using the Variable Infiltration Capacity model, CHIRPS can effectively forecast and analyse hydrologic trend in many African places that suffer from serious droughts [26].

### 3.2. Seasonal Autoregressive Integrated Moving Average (SARIMA)

Box and Jenkins developed a model named Autoregressive Integrated Moving Average (ARIMA) which is used to analyse stationary time-series data sets [16]. This model served efficiently in forecasting non-seasonal data. However, the model seems to be unfit for data with seasonal components. As a consequence, the two scientists developed an extended version of ARIMA, named SARIMA, to deal with monthly or quaternary data that displays annual seasonal patterns.

Generally, seasonal ARIMA models are denoted as follows SARIMA $(p, q, d)$ $(P, Q, D)s$. The structure of the model deals with two features of seasonal time series data. SARIMA $(p, q, d)$ is constructed to show the relationship between consecutive time-series observations, whereas SARIMA $(P, Q, D)s$ is modelled to depict the relationship between corresponding observations of consecutive seasons.

The general equation of the SARIMA $(p, q, d)$ $(P, Q, D)_s$ model is written as follow [27]:

$$\phi_p(B)\Phi_P(B^s)\nabla^d\nabla_S^D z_t = \theta_q(B)\Theta_Q(B^s)a_t \tag{1}$$

where:

$\phi_p$ $(B)$: The non-seasonal autoregressive operator (AR) with $p$th order
$\theta_q$ $(B)$: The non-seasonal moving average operator (MA) with $q$th order
$\Phi_P$ $(B^s)$: The seasonal autoregressive operator with $P$th order

$\Theta^Q$ ($B^s$): The seasonal moving average operator with $Q$th order

$B$: The backshift operator

$\nabla^d$: The non-seasonal differencing with $d$th order

$\nabla_S^D$: The seasonal differencing with $D$th order at s number of lags

$a_t$: an independent variable called also a normal random variable.

The Jenkins and Box methodology was based on several decisive rules to effectively fit the SARIMA model with time-series data using the required parameters for SARIMA $(p, q, d)$ $(P, Q, D)s$ for forecasting future outcomes for a specific period. It is necessary to evaluate the efficiency of the chosen model by estimating the forecasting parameters of the SARIMA model and testing the fitness of the model on the estimated residuals.

Identifying the best model involves the basics of autocorrelation properties. This is the key step in which we can detect suitable(s) model(s) for any specific time-series data. Therefore, we applied Autocorrelation Function (ACF) and Partial Autocorrelation Function (PACF), which consist of identifying the parameter orders of the SARIMA model by measuring the relationship between the current and past observations. Other order selection methods must be taken into consideration, such as Akaike's information criterion (AIC) [28], the Bayesian information criterion (BIC) [29,30], the Ljung–Box test [31,32], and the Jarque–Bera test [33,34]. In addition, different approaches and new testing parameters are applied to evaluate the efficiency of the model and improve the accuracy of forecasting models, such as the likelihood ratio [35,36].

One of the most important steps in forecasting is that time series values must be converted into stationary data. This condition is crucial during the identification process of parameters orders. After achieving stationarity, the time series is suitable for statistic evaluation. For example, selecting the precise order of differencing (d) corresponds to the lowest standard deviation value. After selecting the right orders, taking care to not make the model over or under-differencing, and not getting into root units, evaluating the model adequacy must fulfill statistical characteristics such as mean absolute percentage error (MAPE) or standard error (SE). The best suitable model corresponds to the model that has the lowest MAPE and SE coefficients.

If the statistic laws and the outcomes from residual plot observations suggest that the selected model is not suitable with respect to the historical data, a new proposed model should be identified and operated with respect to all previous checking and validation steps. In some cases, many alternative models can be identified. The best model is the one that has the lowest statistical forecasting errors. The final chosen model would serve in rainfall forecasting. All processing and statistical steps were run in the Spyder environment using (and written in) the Python language.

### 3.3. Statistical Parameters of the SARIMA Model

#### 3.3.1. Term Significance

To prove whether the association between the response and each term in the SARIMA model is statistically significant, we assessed the null hypothesis by comparing the *p*-value for the term to each significance level. The null hypothesis assumed that the term was not significantly different from 0, which denotes no association between the term and the response. Usually, the threshold of significance level (denoted as $\alpha$ or alpha) is around 0.05 which means that a significance level of 0.05 indicates there is a 5% risk that the term is not significantly different from 0, when it is, in fact, significantly different from 0.

#### 3.3.2. Ljung-Box

The Ljung-Box Q statistic is used to test whether a series of observations are random and independent over time. If the observations are not independent, an observation can be correlated with another observation k time units later, establishing a relationship called autocorrelation. Autocorrelation can affect the accuracy of a time-based forecasting model, such as a time series plot, and lead to misinterpretation of data.

### 3.3.3. Heteroscedasticity

Heteroskedasticity is used to evaluate standardized residuals. It checks whether the sum-of-squares in the first third of the sample is significantly different than the sum-of-squares in the last third of the sample. The null hypothesis assumes no heteroskedasticity. In the context of regression modeling, heteroscedasticity means that the conditional variance of the data is not constant. Conditional variance is the variability of the dependent variable (named y) for each value of time period t (in case of time series data), or each value of explanatory variables (named X) in general.

### 3.3.4. Jarque-Bera

The Jarque-Bera test is used to assess the normality of standardized residuals. It is generally a goodness-of-fit test of whether time series data have kurtosis and skewness matching a normal distribution. The resultant value from the test is always non-negative. The null hypothesis assumes the normality of the sample data. In a case where the test statistic is far from zero, it presumes that the data do not have a normal distribution.

### 3.4. Standard Precipitation Index (SPI)

Over the years, many drought indices have been developed and used around the world. The standard precipitation index (SPI) is a powerful index. It is based on precipitation as parameter to analyse rainfall deficits and their potential impacts. It is also very effective in analyzing both wet and dry periods. The reason for using SPI is that computing of the parameters deals only with precipitation, which could be useful in case of lack of supplementary data (temperature and humidity, among others). In addition, it is efficient in characterizing drought (or abnormal wetness) at various time scales depending on the time availability of water resources type (e.g., snowpack, groundwater, soil moisture, reservoir storage and river discharge). Another distinctive feature of the SPI index is that it is more comparable across regions with different climates than other indexes (like Palmer Severity Drought Index (PDSI)). In addition, computing of SPI is less complex than other indexes (PDSI). It generally uses long-term precipitation data which are fitted on a probability distribution (such as gamma distribution) and then transformed into a normal distribution so that the generated SPI value is zero. Generated values can be positive or negative, where positive amounts are greater than the median precipitation and negative rates indicate less than the median precipitation. The interpretation of SPI values is shown in the following Table 1 [37]:

**Table 1.** Standard Precipitation Index classes and interpretation.

| SPI Values | Interpretation |
|:---:|:---:|
| $\geq 2$ | Extremely wet |
| [1.5 to 1.99] | Very wet |
| [1.0 to 1.49] | Moderately wet |
| [−0.99 to 0.99] | Near normal |
| [−1.0 to −1.49] | Moderately dry |
| [−1.5 to −1.99] | Severely dry |
| $\leq -2$ | Extremely dry |

## 4. Results and Discussion

This section describes the use of several Python packages in the study. The tasks of reading a CSV file and data analysis were performed using the "pandas" package. The statistical analysis and time-series forecasting were carried out using the "SARIMAX" function of the "statsmodels" package. Data visualization was performed using the "Matplotlib" package. The time-series data were processed in Python to remove seasonal variation

and make them stationary. The initial statistical evaluation indicated that rainfall has a fluctuating statistical property that depends on time. The PACF plot (Figure 3A) reveals that the series has a strong positive autocorrelation for a high number of lags, requiring a differencing order "d" to ensure stationarity in the time-series data. The optimal differencing order was determined to be "d = 1", where the standard deviation (SD) was the lowest (SD = 1.70–15 for d = 1, and SD = 2.26–15 for d = 2). The low SD value indicates that the data points are close to the mean. Upon inspection of the ACF and PACF plots of the differenced series (Figure 3B), adding AR (1) and MA (1) terms to the model was necessary to adjust the sharp cutoff in the series, as the time-series data appeared to be under-differenced. The positive aspect of the first observation (lag) in the PACF supported the addition of the AR (1) term. At this stage (Figure 3D), ACF and PACF plots still show a consistent and strong seasonal pattern, which repeatedly arise in the 11th, 12th, 24th, and 25th lags. Therefore, we added one seasonal difference "D" to the series. Based on the last observation of the ACF and PACF plot (Figure 3E), we detected a persistent negative autocorrelation at the 11th lag which repeatedly appeared with less amplitude. Consequently, this point made us consider adding an SMA (1) term "Q". The final ACF and PACF plots of the deduced SARIMA can be seen in Figure 3F. According to this, we can conclude that the best SARIMA for this time-series data is SARIMA (1,1,1) (0,1,1)12. A summary of the model is illustrated in Table 2.

**Table 2.** Table summary for SARIMA (1,1,1) (0,1,1)12 model.

| SARIMA Results | | | | | | |
|---|---|---|---|---|---|---|
| Dep. Variable | | | Rainfall | | | |
| No. of Observations | | | 468 | | | |
| Model | | | SARIMA (1, 1, 1) × (0, 1, 1, 12) | | | |
| Log Likelihood | | | −2061.268 | | | |
| AIC | | | 4130.536 | | | |
| BIC | | | 4147.017 | | | |
| HQIC | | | 4137.029 | | | |
| Sample | | | 01-01-1981–12-01-2019 | | | |
| Covariance Type | | | opg | | | |
| | coef | std err | z | $p > |z|$ | [0.025 | 0.975] |
| ar. L1 | 0.0586 | 0.043 | 1.349 | 0.177 | −0.026 | 0.144 |
| ma. L1 | −0.9963 | 0.035 | −28.824 | 0 | −1.064 | −0.929 |
| ma. S. L12 | −0.9973 | 0.54 | −1.847 | 0.065 | −2.056 | 0.061 |
| sigma2 | 449.4187 | 234.936 | 1.913 | 0.056 | −11.047 | 909.884 |
| Ljung-Box (Q) | 48.19 | | Jarque-Bera (JB) | | 393.37 | |
| Prob(Q) | 0.18 | | Prob (JB) | | 0 | |
| Heteroskedasticity (H) | 0.94 | | Skew | | 1.21 | |
| Prob(H) (two-sided) | 0.68 | | Kurtosis | | 6.86 | |

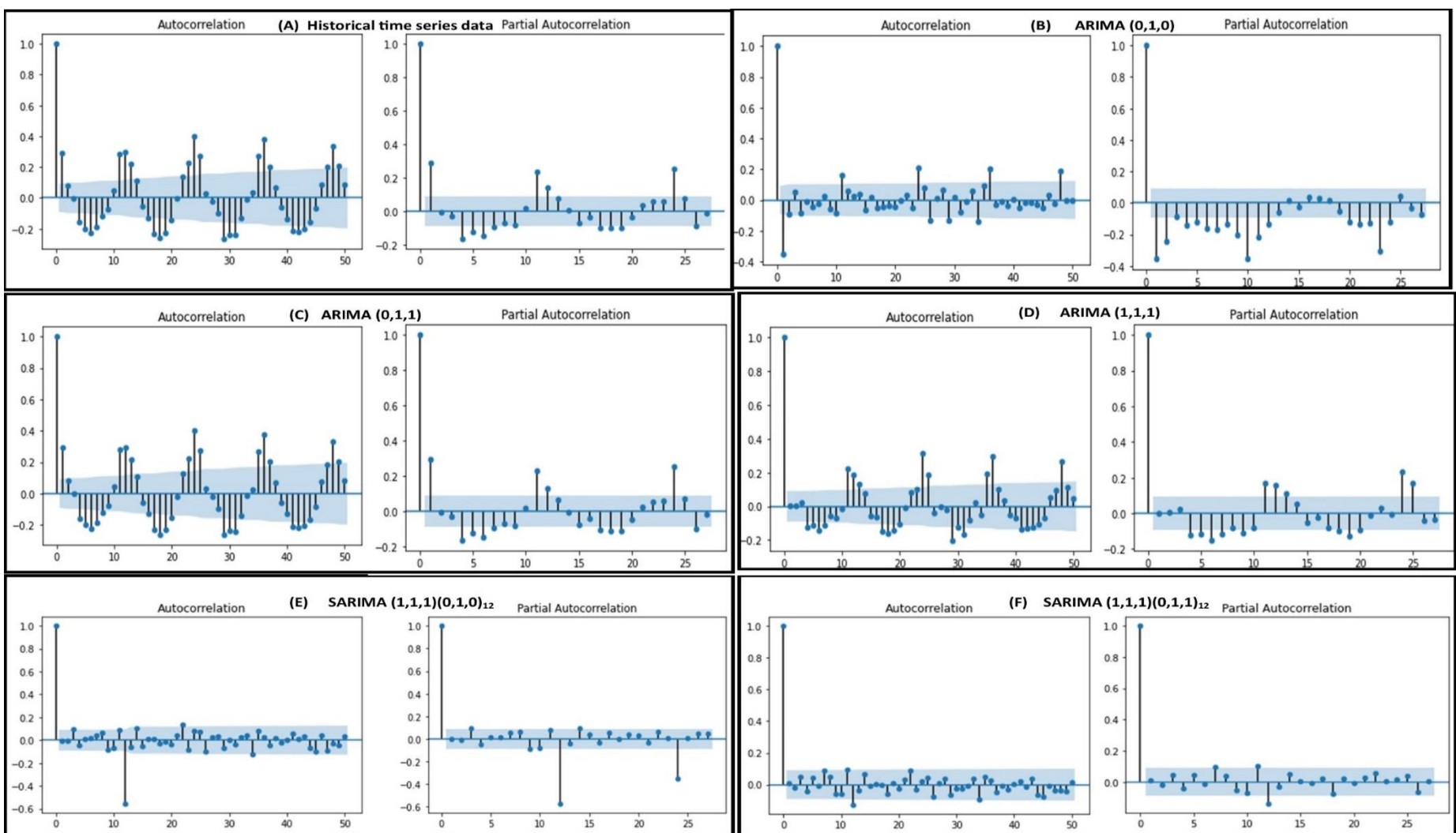

**Figure 3.** ACF and PACF plots for different ARIMA and SARIMA models.

### 4.1. Forecasting Accuracy Measurement

The best way to estimate and compare forecasted with actual values is by operating backward forecasting. The SARIMA predictive values were calculated and compared with the historical values of the time series data. Many criteria and statistical tools can evaluate forecasting accuracy. The Mean absolute error (*MAE*) is one of the most common tools for evaluating forecasting performance. The equation is formulated as follows [38]:

$$MAE = \frac{\sum_{i=1}^{n}|y_i - x_i|}{n} \tag{2}$$

where $y_i$ is the forecasted value, and $x_i$ is the observed value.

The *MAE* value is expressed as a percentage (%). The lower the value, the better the forecasting. Compared to other models, it seems that the chosen SARIMA (1,1,1) (0,1,1)12 model generated the lowest *MAE* value, which was 15.9%. Hence, the evaluation of the forecasting accuracy of this SARIMA model accords with the criteria of Lewis (1982) [39], which classifies our model with a good forecasting accuracy.

### 4.2. Summary of the SARIMA Model

The SARIMA (1,1,1) (0,1,1)12 model was found to have the best performance statistically compared to other models. It had the lowest standard error of residual variance ($\sigma^2$) compared to the others. However, the value was still high, indicating that the forecasted model was not a perfect fit to the observed data. This can be seen in Figure 4, which shows the heterogeneity between the forecasted and observed values.

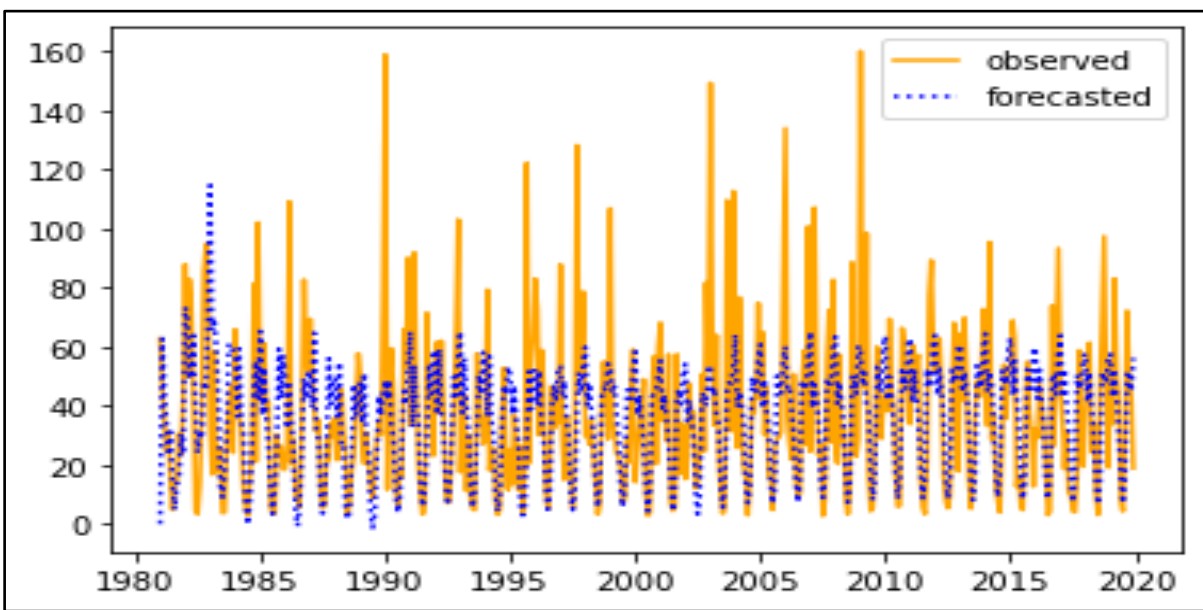

**Figure 4.** Comparison between forecasted and observed data.

The log-likelihood ratio (log likelihood) measures how well the forecasted model fits the observed values. A higher ratio indicates a better fit. Our selected SARIMA model had the highest log likelihood ratio compared to the other models. AIC (Akaike information criterion), BIC (Bayesian information criterion), and HQIC (Hannan-Quinn information criterion) are statistical estimators that indicate how well the forecasted model fits the observed data. Models with the lowest AIC, BIC, and HQIC values are considered to have the best fit. However, it is important to note that the AIC ratio may be penalized if the corresponding model uses multiple parameters, as is the case with our SARIMA model.

Overall, compared to other models, our chosen SARIMA model had the highest log likelihood ratio and the lowest AIC, BIC, and HQIC values. Hence, it can be concluded that our model was the best fit for forecasting values (as shown in Table 3).

**Table 3.** Statistical assessment between different ARIMA and SARIMA models.

|  | SARIMA (1,1,1) $(0,1,1)_{12}$ | SARIMA (0,1,1) $(0,1,1)_{12}$ | SARIMA (1,1,1) $(0,1,0)_{12}$ | ARIMA (0,1,1) | ARIMA (1,1,1) |
|---|---|---|---|---|---|
| *MAE* (%) | 15.91 | 15.98 | 22.25 | 43.8 | 40.8 |
| Log. Likelihood | −2061.268 | −2062.039 | −2224.694 | −2203.468 | −2182.855 |
| AIC | 4130.078 | 4130.536 | 4455.387 | 4412.935 | 4373.710 |
| BIC | 4142.439 | 4147.017 | 4467.748 | 4425.374 | 4390.296 |
| HQIC | 4134.948 | 4137.029 | 4460.257 | 4417.831 | 4380.237 |
| SE ($\sigma^2$) | 234.936 | 432.148 | 1009.043 | *** | *** |
| SE | 0.618 | 0.996 | 1.034 | 1.25 | 1.2 |

*** mean that this statistical variable is not available/ relevant for ARIMA model.

### 4.2.1. Term Significance

The statical significance of the model parameters is supported by the probability values (*p*-values). The null hypothesis regarding this probability value is that each parameter of our chosen model is not statistically significant. In this case, the *p*-values of all parameters, except AR (1) were equal or less than 0.05 (critical value) which allows us to reject the null hypothesis and deduce that all cited parameters are statically significant. As for AR (1), the *p*-value was higher than 0.05 (0.177 > 0.05). In this case, we conclude that the null hypothesis is retained but this does not allow us to say that the AR (1) is not statically dependent [40]. This is shown in Figure 3 where we can see the difference of the ACF and PACF plots when we add the AR (1) term to the model (plot D). However, we want to evaluate the accuracy performance of the SARIMA model by removing the AR (1) parameter. According to the results in Table 3, although there is a big similarity in both models, we can see that the standard errors (SE) have deteriorated, which shows that the accuracy of the SARIMA (0,1,1) (0,1,1)12 has declined compared to the previous model. This supports our assumption that the AR (1) term has a statistical significance, but could not as indicated by the *p*-value.

### 4.2.2. Ljung-Box

According to E.P Box et al. (1970) [16,27,33], this test is generally applied to evaluate the randomness of data by checking if (or not) the autocorrelations of time series data are different from zero. Therefore, the Ljung-box ratio is set to test a null hypothesis that assumes that the autocorrelations are equal to zero up to a certain lag n. Since our probability (0.18) was above the specified critical value (0.05), we cannot reject the null hypothesis. Therefore, we can say that the autocorrelations of one or more lags are different from zero as shown in the ACF and PACF plots (Figure 3). Accordingly, our time-series data are random and independent over time. Autocorrelations of one to more lags can be different from zero.

### 4.2.3. Heteroscedasticity

The heteroscedasticity statistical test is used to evaluate if the residual errors have the same (constant) variance (homoscedastic). The related probability value is compared to the critical value in a way to accept or reject the null hypothesis. Since the resultant probability value (0.66) was higher than 0.05, we accept the null hypothesis that the residuals have a constant variance. It seems that this point is quite reasonable due to the fact that the historical data show a close resemblance in some specific years. The data show the same precipitation amounts (close monthly or yearly average) in some years. Therefore, there may be the same residual variance for some lags.

4.2.4. Jarque-Bera

This statistical test of Jarque and Bera (1980) [41] determines if the data have a normal distribution. The correspondent probability value tests the normality of the errors. Since our *p*-value (0) was under 0.05, we reject the null hypothesis that the data are normally distributed. Since it has a seasonal variation, our data series must not be normally distributed. The skewness and kurtosis are correspondent features of the Jarque-Bera test. Therefore, we conclude that our time series data have a slight positive skew and a large kurtosis.

4.2.5. Forecasting Results

The SARIMA (1,1,1) (0,1,1)12 forecast trend is depicted in Figure 5. The model shows a consistent precipitation trend with an average of 470 mm/year. The highest amounts of precipitation are predicted for January (60 mm) and December (55 mm), while the lowest are expected in July (7 mm). However, the model's forecast deviates from the observed historical data, as it fails to capture irregular precipitation patterns where monthly rainfall exceeds 100 mm. The model projects similar precipitation behavior over a five-year period, as it mimics the first-year trend and repeats it for the following years. This is due to SARIMA's one-period-ahead prediction approach, which assumes previously forecasted data as historical data.

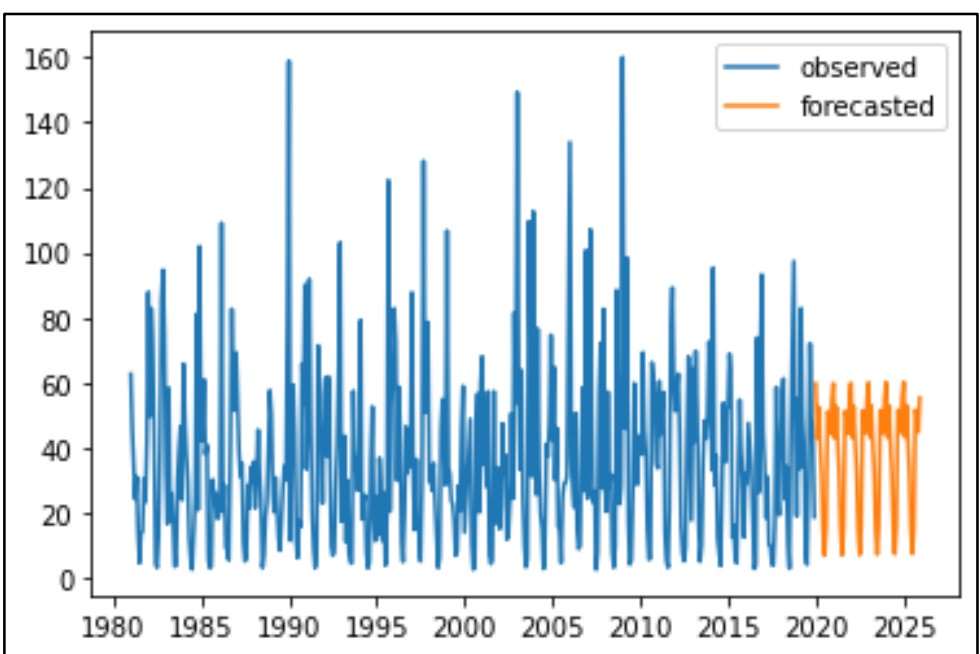

**Figure 5.** Rainfalls forecasting trend from 2020 to 2025.

SARIMA models are favored for their ease of implementation and accuracy in short-term forecasting. It has been demonstrated that SARIMA can provide a sufficient modeling approach for weather forecasting. Depending on the data type, SARIMA can generate multiple potential forecasting models [18,42,43], but generating one SARIMA model is also feasible. Our case study is similar to the study conducted by Dimri et al. (2020), where the authors found that SARIMA (0,1,1) (0,1,1)12 was the most appropriate model for precipitation forecasting in Uttarakhand, India.

SARIMA is not ideal for long-term forecasting, as the model relies on a forecasting equation for one period ahead and repeats it for future periods as desired. Over a long-term period, this extrapolation can be rigid, especially if the model produces the best forecasting equation. It would be more beneficial to combine SARIMA models with other sources and technologies that enhance multi-period forecasting [13].

Additionally, SARIMA requires a substantial amount of data, with a minimum of 50 values, and optimally around 100 values [43]. However, it may be difficult to obtain this amount of data in some cases due to uncertainties or a lack of measurement tools. New forecasting technologies typically use a small amount of data over a shorter period.

### 4.2.6. Drought Forecasting

Based on the results generated from Figure 6 and Tables 4 and 5, it was determined that the region is at high risk of drought. The results were based on the monthly variation of the SPI index. The overall evaluation shows that around 50% of the period is classified as near normal, with a significant dry stage constituting 35% of the whole period. The results also show that the area experiences a rare wet period that covers nearly 15% of the estimated period. The refined classification allows for the distinction of different drought categories. It confirms that the main drought category is near normal. On the other hand, the dry period classification reveals a severe drought intensity period, where 26% of the period is categorized as very to extremely dry (13% each), and only 9% of the cumulative period is defined as moderately dry. The wet period is classified with 6.5% of the total period as moderately wet, 5.5% as very wet, and only 2% as extremely wet.

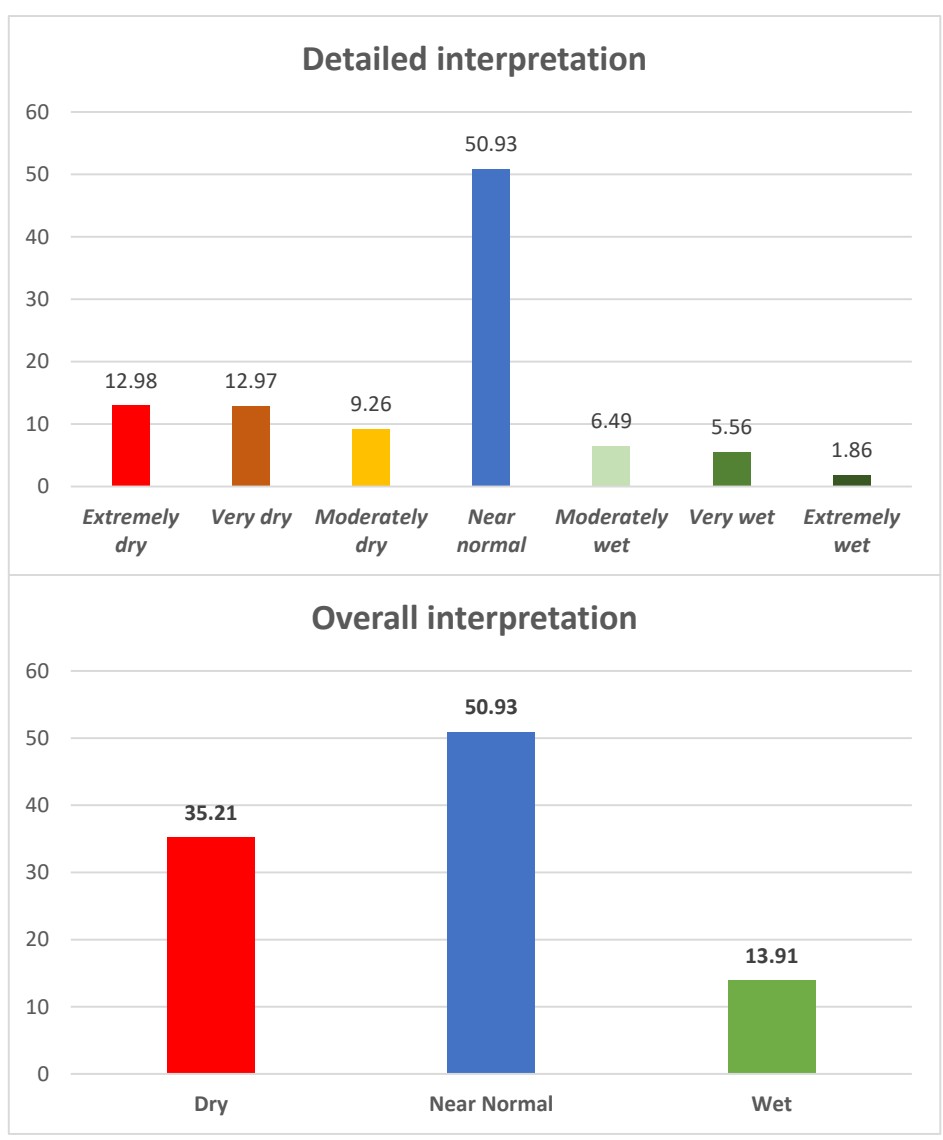

**Figure 6.** Drought frequency distribution based on monthly forecasted SPI from 2022 to 2030.

**Table 4.** SPI prediction from 2022 to 2030.

| Month | SPI | Month | SPI | Month | SPI |
|---|---|---|---|---|---|
| Jan-22 | −2.9 | May-25 | 0.54 | Sep-28 | −1.96 |
| Feb-22 | −2.81 | Jun-25 | 0.23 | Oct-28 | −1.92 |
| Mar-22 | −2.83 | Jul-25 | 0.39 | Nov-28 | −1.91 |
| Apr-22 | 2.82 | Aug-25 | 0.39 | Dec-28 | −2.01 |
| May-22 | −2.78 | Sep-25 | 0.39 | Jan-29 | 0.08 |
| Jun-22 | 2.03 | Oct-25 | 0.39 | Feb-29 | −1.31 |
| Jul-22 | 1.56 | Nov-25 | 0.39 | Mar-29 | 0.27 |
| Aug-22 | 1.55 | Dec-25 | 0.39 | Apr-29 | −0.53 |
| Sep-22 | 1.55 | Jan-26 | 0.4 | May-29 | −0.88 |
| Oct-22 | 1.55 | Feb-26 | 0.42 | Jun-29 | −1.34 |
| Nov-22 | 1.55 | Mar-26 | 0.37 | Jul-29 | −1.16 |
| Dec-22 | 1.55 | Apr-26 | −0.45 | Aug-29 | −1.56 |
| Jan-23 | 0.73 | May-26 | 0.4 | Sep-29 | −2.12 |
| Feb-23 | 0.96 | Jun-26 | −0.84 | Oct-29 | −2.09 |
| Mar-23 | 0.46 | Jul-26 | −0.01 | Nov-29 | −2.08 |
| Apr-23 | −0.38 | Aug-26 | −1.07 | Dec-29 | −2.17 |
| May-23 | 0.8 | Sep-26 | −1.81 | Jan-30 | −1.18 |
| Jun-23 | 0.87 | Oct-26 | −1.77 | Feb-30 | −1.34 |
| Jul-23 | 1.15 | Nov-26 | −1.76 | Mar-30 | 0.24 |
| Aug-23 | 1.16 | Dec-26 | −1.87 | Apr-30 | −0.56 |
| Sep-23 | 1.16 | Jan-27 | 0.29 | May-30 | −0.9 |
| Oct-23 | 1.16 | Feb-27 | 0.24 | Jun-30 | −1.58 |
| Nov-23 | 1.16 | Mar-27 | 0.34 | Jul-30 | −1.56 |
| Dec-23 | 1.16 | Apr-27 | −0.48 | Aug-30 | −1.84 |
| Jan-24 | 0.62 | May-27 | 0.27 | Sep-30 | −2.33 |
| Feb-24 | 0.78 | Jun-27 | −0.95 | Oct-30 | −2.3 |
| Mar-24 | 0.43 | Jul-27 | −0.39 | Nov-30 | −2.3 |
| Apr-24 | −0.4 | Aug-27 | −1.15 | Dec-30 | −2.38 |
| May-24 | 0.67 | Sep-27 | −1.85 | | |
| Jun-24 | 0.55 | Oct-27 | −1.81 | | |
| Jul-24 | 0.77 | Nov-27 | −1.81 | | |
| Aug-24 | 0.77 | Dec-27 | −1.91 | | |
| Sep-24 | 0.77 | Jan-28 | 0.18 | | |
| Oct-24 | 0.77 | Feb-28 | 0.08 | | |
| Nov-24 | 0.77 | Mar-28 | 0.31 | | |
| Dec-24 | 0.77 | Apr-28 | −0.51 | | |
| Jan-25 | 0.51 | May-28 | 0.14 | | |
| Feb-25 | 0.6 | Jun-28 | −1.12 | | |
| Mar-25 | 0.4 | Jul-28 | −0.77 | | |
| Apr-25 | −0.43 | Aug-28 | −1.32 | | |

**Table 5.** Frequency of forecasted Monthly SPI values from 2022 to 2030.

| SPI | Frequency | Percentage | Interpretation | Cumulative Percentage |
|---|---|---|---|---|
| −2.9 | 1 | 0.93 | | |
| −2.8 | 3 | 2.78 | | |
| −2.4 | 1 | 0.93 | | |
| −2.3 | 3 | 2.78 | Extremely dry | |
| −2.2 | 1 | 0.93 | | |
| −2.1 | 3 | 2.78 | | |
| −2 | 2 | 1.85 | | 12.98 |
| −1.9 | 5 | 4.63 | | |
| −1.8 | 6 | 5.56 | Very dry | |
| −1.6 | 3 | 2.78 | | 12.97 |
| −1.3 | 4 | 3.7 | | |
| −1.2 | 3 | 2.78 | Moderately dry | |
| −1.1 | 2 | 1.85 | | |
| −1 | 1 | 0.93 | | 9.26 |
| −0.9 | 2 | 1.85 | | |
| −0.8 | 2 | 1.85 | | |
| −0.6 | 1 | 0.93 | | |
| −0.5 | 4 | 3.7 | | |
| −0.4 | 4 | 3.7 | | |
| 0 | 1 | 0.93 | | |
| 0.1 | 3 | 2.78 | | |
| 0.2 | 4 | 3.7 | Near normal | |
| 0.3 | 5 | 4.63 | | |
| 0.4 | 12 | 11.11 | | |
| 0.5 | 3 | 2.78 | | |
| 0.6 | 3 | 2.78 | | |
| 0.7 | 2 | 1.85 | | |
| 0.8 | 8 | 7.41 | | |
| 0.9 | 1 | 0.93 | | 50.93 |
| 1 | 1 | 0.93 | | |
| 1.1 | 1 | 0.93 | Moderately wet | |
| 1.2 | 5 | 4.63 | | 6.49 |
| 1.5 | 5 | 4.63 | | |
| 1.6 | 1 | 0.93 | Very wet | 5.56 |
| 2 | 1 | 0.93 | | |
| 2.8 | 1 | 0.93 | Extremely wet | 1.86 |

## 5. Conclusions

We estimated the amount of precipitation in the semi-arid city of Zaghouan. After making the time-series data stationary, SARIMA was used to find the most appropriate parameters for the model (SARIMA (1,1,1) (0,1,1)12). SARIMA (1,1,1) (0,1,1)12 showed the best performance according to statistical parameters, generating a precipitation trend similar to historical tendencies. The MAPE showed that SARIMA (1,1,1) (0,1,1)12 can also accurately predict drought events based on precipitation data at different time scales. However, additional climatic parameters, such as temperature, wind, and relative humidity, should be considered for drought forecasting. The prediction of SPI was also calculated from 2022 to 2030, and the evaluation of the forecasted monthly data showed a "near normal" trend with significant dry events and rare wet months, which make up approximately 13% of the total forecasted months. The results are presented in Tables 4 and 5.

Drought forecasting is crucial for drought management and plays a vital role in providing early warning of potential drought events, which can help preserve crop yields and quality in regions with degraded land that hinders land use sustainability.

**Author Contributions:** Conceptualization, O.W.; methodology, O.W. and M.B.; software, O.W.; validation, O.W., M.B. and M.-M.S.; formal analysis, O.W.; resources, M.B.; writing—original draft preparation, O.W.; writing—review and editing, M.B. And M.-M.S.; visualization, O.W. All authors have read and agreed to the published version of the manuscript.

**Funding:** This research received no external funding.

**Data Availability Statement:** Available rainfall data can be found in CHRIPS Website via this link: https://www.chc.ucsb.edu/data/chirps (accessed on 1 February 2023).

**Conflicts of Interest:** The authors did not receive support from any organization for the submitted work. The authors have no relevant financial or non-financial interests to disclose.

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
