# Peer review of "Precipitation Forecasting and Monitoring in Degraded Land: A Study Case in Zaghouan"

_land, doi:10.3390/land12040738_

Round 1
Reviewer 1 Report
Modifications are described in the attached document.
Please also review the spell and style check in methodology part.
Thanks for contributing to the Land journal.

Author Response
Peer Review Office
Land-MDPI
January 31h, 2023
AUTHORS’ RESPONSES TO THE REVIEWERS’ COMMENTS
Dear Editor,
We would like to thank you for offering us the opportunity to prepare a complete manuscript for publication once a final revision considering the submitted manuscript has been achieved. We would like to thank the reviewers as well for their serious reading and valuable comments that have really helped us improve the final version of our manuscript.
We were pleased that all reviewers considered the manuscript an important contribution to the field and worth publishing after some major corrections. The reviewers’ comments and suggestions were taken into consideration in order to improve the clarity of the manuscript. We followed all of their recommendations and suggestions and updated our manuscript accordingly. We believe that the revised version is a much more focused and stronger paper. We have answered the comments of each reviewer individually. A brief response list of answers to their comments is provided accordingly.
We sincerely hope that you will find our revised manuscript worth publishing in your prestigious journal. We look forward to receiving a positive reply
,
Kind regards,
Okba Weslati
The corresponding author

Reviewer 2 Report
January 20, 2023
Manuscript: Precipitation forecasting and monitoring in degraded land; A study case in Zaghouan
The manuscript studies precipitation forecasting and monitoring in degraded land using the best Seasonal Autoregressive Integrated Moving Average (SARIMA) model on ground-based monthly precipitation data in the northeast of Tunisia. The article is well-organized and written, with enough scientific content. However, some revisions are required for publication.
The title of the manuscript accurately reflects the content of the article. The abstract provides a complete picture of the manuscript content and is concise. The keywords are appropriately reflective of the manuscript content. In line 15, it is necessary to indicate the drought index used in this study.
The introduction provides enough background on using SARIMA models to forecast seasonal time series data, but it should indicate the novelty of the study. Lines 31-34 should add more information about droughts, including the types of droughts (i.e., meteorological, agricultural, hydrological) and their main characteristics (i.e., start, end, duration, intensity, and severity). Also, more information about the main drivers of land degradation dominant in Zaghouan is needed in line 108.
The narrative of the study area section is clear and concise. To improve the clarity to the reader, it is suggested to add coordinate axes to both maps in Figure 1, line 145. Lines 149-150 should provide more details about the rain gauge's location (Figure 1). It is also important to mention whether CHIRPS-based precipitation or rain gauge-based data has been used and briefly explain it.
The software used for processing should be mentioned in lines 255-256. Also, in lines 258-269, the criteria behind choosing the standard precipitation index (SPI) should be briefly explained.
The results and discussion section are clear and concise, but some suggestions have been made. Lines 274-284 should be moved to the Study Area section. In lines 286-287, the Python packages used should be listed. For clarity, on page 12, lines 40, 57, 67, 76, and 84, the meaning of these statistical parameters should be briefly explained in the methodology section.
The conclusions section is straightforward and harmonized with the previous sections.
Overall, the manuscript is a valuable contribution to the field of precipitation forecasting and monitoring in degraded land.

Author Response
Peer Review Office
Land-MDPI
January 31th, 2023
AUTHORS’ RESPONSES TO THE REVIEWERS’ COMMENTS
Dear Editor,
We would like to thank you for offering us the opportunity to prepare a complete manuscript for publication once a final revision considering the submitted manuscript has been achieved. We would like to thank the reviewers as well for their serious reading and valuable comments that have really helped us improve the final version of our manuscript.
We were pleased that all reviewers considered the manuscript an important contribution to the field and worth publishing after some major corrections. The reviewers’ comments and suggestions were taken into consideration in order to improve the clarity of the manuscript. We followed all of their recommendations and suggestions and updated our manuscript accordingly. We believe that the revised version is a much more focused and stronger paper. We have answered the comments of each reviewer individually. A brief response list of answers to their comments is provided accordingly.
We sincerely hope that you will find our revised manuscript worth publishing in your prestigious journal. We look forward to receiving a positive reply
,
Kind regards,
Okba Weslati
The corresponding author

Reviewer 3 Report
Dear Authors,
I made a strong review of your article and found that there are a number of corrections to be made in your article. Overall, I inserted 60 corrections in my review of your article. Please find herewith attached my review of your article with all these corrections.
Also, please note that your reference list is not in the format of this journal. From the beginning of this article, when you are providing the first reference, the reference should be given as number (1). Afterwards all the references in this article should be numbered till the end. Then in this reference list you should give number 1 to the last number.

Author Response

(The authors gave the same response as above.)

Round 2
Reviewer 3 Report
Dear Authors,
In your revised version, you provided the reference numbers.
The first reference you cited in your revised version was 7, which is given on page 2. Before this citation, you did not insert the reference numbers 1 to 6. Please make this correction.
On page 3, you cited references 16 and 18. I did not find reference number 17 before reference 18. Please insert reference 17.
On page 4, you cited Figure 2, but did not insert the figure on this page or the next page. I found Figure 2 only on page 9. Please move Figure 2 to page 5.
Please make all the corrections given above and submit a new revised version of your paper.
Best wishes,